

# Evaluations of an ocean bottom electro-magnetometer and preliminary results offshore NE Taiwan

**Ching-Ren Lin[1], Chih-Wen Chiang[2], Kuei-Yi Huang[2], Yu-Hung Hsiao[3], Po-Chi Chen[3], Hsu-Kuang Chang[3], Jia-Pu Jang[3], Kun-Hui Chang[1], Feng-Sheng Lin[1], Saulwood Lin[4], and Ban-Yuan Kuo[1]**

Institute of Earth Sciences, Academia Sinica, Taipei 11529, Taiwan, R.O.C
Institute of Earth Sciences, National Taiwan Ocean University. Keelung 20224, Taiwan, ROC.
Taiwan Ocean Research Institute, National Applied Research Laboratories, Kaohsiung 80143, Taiwan, R.O.C.
Institute of Oceanography, National Taiwan University, Taipei 10617, Taiwan, R.O.C

Corresponding author: Chih-Wen Chiang

Address: Institute of Earth Sciences, National Taiwan Ocean University, Keelung 20224, Taiwan

E-mail: zjiang@ntou.edu.tw

Tel: +886-2-24622192 Ext.6513

Fax: +886-2-24625038

April, 2019



1                 **ABSTRACT**

The first stage of field experiments involving the design and construction of a low-
power consumption ocean bottom electro-magnetometer (OBEM) has been completed.
To improve the performance of the OBEM, we rigorously evaluated each of its units,
e.g., the data loggers, acoustic parts, internal wirings, and magnetic and electric sensors,
to eliminate unwanted events such as unrecovered or incomplete data. The evaluations
of the procedure included the following.
• Data logger: digitizer sensitivity, linearity, and errors
• Acoustic transceiver: "ENABLE," "DISABLE," "RANGE," "RELEASE1,"
"RELEASE2," and "OPTION1" functions
• Magnetic sensor: sensitivity of the fluxgate and orthogonality
• Electrical receiver: potential voltage, impedance, and frequency responses
• Power consumption: the maximum operating current of two sets of batteries
• Deployment and recovery procedures on deck
We confirmed the optimal performance of the OBEM after repeatedly testing the
procedures.
The first offshore deployment of the OBEM together with ocean bottom seismographs
(OBSs) was performed in NE Taiwan, where the water depth is approximately 1,400
m. The total intensity of the magnetic field (TMF) measured by the OBEM varied in
the range of 44,100–44,150 nT, which corresponded to the proton magnetometer
measurements. The daily variations of the magnetic field were recorded using the two
horizontal components of the OBEM magnetic sensor. We found that the inclinations
and magnetic data of the OBEM varied with two observed earthquakes when compared
to the OBS data. The potential fields of the OBEM were slightly, but not obviously,
affected by the earthquakes.
Keywords: OBEM; data logger; acoustic transceiver; fluxgate; non-polarizing
electrodes.
**1. Introduction**
Marine electromagnetic exploration is a geophysical prospecting technique used to



reveal the electrical resistivity features of the oceanic upper mantle down to depths of
several hundreds of kilometers in different geologic and tectonic environments, such as
in areas around mid-oceanic ridges, areas around hot-spot volcanoes, subduction zones,
and normal ocean areas between mid-oceanic ridges and subduction zones zones (Ellis
et al., 2008; Evans et al., 2005; Key, 2012; Utada, 2015). Marine controlled source
electromagnetic (MCSEM) methods have been used for methane hydrate mapping to
detect offshore hydrocarbons (Constable, 2010; Goto et al., 2008; Schwalenberg et al.,
2017; Weitemeyer et al., 2011; Weitemeyer et al., 2006).

Even though many magnetotelluric explorations have investigated deep electrical
structures on Taiwan (Bertrand et al., 2009; Bertrand et al., 2012; Chiang et al., 2011a;
Chiang et al., 2010; Chiang et al., 2015; Chiang et al., 2008), there were no marine
electromagnetic experiments around Taiwan until 2010. The first MCSEM survey was
carried out for gas hydrate investigations offshore SW Taiwan (Hsu et al., 2014).
Marine electromagnetic methods have gradually gained the attention of Taiwanese
scientists following these MCSEM experiments (Chiang et al., 2012; Chiang et al.,
2011b).

The first generation of ocean bottom seismographs (OBSs) was developed by the
Institute of Earth Sciences, Academia Sinica (IES), Taiwan Ocean Institute, National
Applied Research Laboratories, and the Institute of Undersea Technology, National Sun
Yat-sen University (OBS R&D team), in 2009, the so-called Yardbird-20s. These OBSs
have acquired large amounts of data via a series of deployments offshore Taiwan that
can be used to study plate tectonics and crustal characteristics (Kuo et al., 2015; Kuo et
al., 2012; Kuo et al., 2014). Subsequently, the OBS R&D team developed an ocean
bottom electro-magnetometer (OBEM) modified from the OBS based on important
developmental experiments.

The novel OBEM was constructed by the OBS R&D team and has completed the first
stage of field experiments by the Institute of Earth Sciences, National Ocean Taiwan
University, and IES. One OBEM and six broadband OBSs, so-called BBYBs, were
deployed at the western end of the Okinawa Trough (OT), NE Taiwan, for field testing
in March 2018. The water depth in this area is approximately 1,400 m. All the



instruments were successfully recovered in May 2018 after collecting the first OBEM
field data in Taiwan. Here, we introduce the OBEM design, specifications, calibration
procedures, and its further developments and improvements.

**2.  The OBEM design**
The OBEM is designed to be wireless deep-underwater equipment; however, the power
supply is limited for the wireless OBEM because the batteries cannot be directly
charged via electric cables from vessels. Therefore, designing low-power consumption
for the OBEM and high-efficiency battery packs is critically required for long periods
of operation. The major units of the OBEM include a data logger, a magnetic sensor, a
tiltmeter, electric receivers with an arm-folding mechanism, a relocation system,
recovery units, and an anchor. All the units for the OBEM use nonmagnetic materials
(e.g., the screws and anchor). Figure 1 shows a block diagram of the OBEM. We
designed the data logger, release mechanism, and the OBEM platform to integrate all
the sensors or units purchased from related manufactories and focused on the issues of
saving power and reducing costs. The detailed requirements of the OBEM are listed
below.
1. A magnetic sensor with three axes for measuring magnetic fields
2. A tiltmeter with two axes for measuring leveling changes to correct the tilt error

of the magnetic sensor

3. Two pairs of non-polarized electrodes with 2-m bendable arms with a total

distance between the electrodes of approximately 4.5 m

4. A highly accurate data logger with at least seven channels and a sampling rate

of greater than or equal to 10 samples per second (SPS)

5. An operation time of more than 90 days
6. An internal timing error of less than 3 s y$^{-1}$ synchronized with GPS
7. Acoustic relocation and recovery control systems
8. A power consumption of less than 1.5 W
9. A radio beacon, flush beacon, reflect label, and orange flag for identification on

the sea surface during instrument recovery

10. A 0.75 m s$^{-1}$ subside rate for deployment and float up rate for recovery
11. A maximum deployment depth of more than 6,000 m appropriate for most

seawater depths offshore Taiwan




The solutions found for the OBEMs are listed below.
1. A fluxgate with three axes with a sensitivity of ±70,000 nT
2. A tiltmeter with two axes with inclinations of ±30°
3. Two pairs of silver chloride electrodes with a 2-m arm-folding mechanism
4. A low noise and low-power consumption eight differential channel 24-bit A/D

data logger with an accurate internal timing clock

5. Acoustic transponder and controller units
6. Radio beacon and flash beacon units
7. An OBEM platform modified from that of OBS
8. High-efficiency lithium battery packs for the sensors and data logger

**3.  Units of the OBEM and their specifications**
The OBEM is recovered by releasing its anchor from the seafloor via an on-board
acoustic command. The OBEM is returned to the sea surface via buoyancy when the
anchor is released. There are two typical release mechanisms available for OBEMs to
unlock their anchors: spin motor and burn-wire systems (Kasaya and Goto, 2009). The
OBEM uses the burn-wire system because it weighs less than the spin motor system.
The acoustic controller and transducer use ORE #B980175 ASSY PCB and #D980709,
respectively, manufactured by EdgeTech, USA, for the corresponding functions of
OBEM recovery and underwater ranging. The ASSY PCB acoustic controller uses a
binary FSK encoder, including the commands "RELEASE1," "RELEASE2,"
"DISABLE," "ENABLE," and "OPTIONAL1." The frequency of the acoustic range
ranges from 7.5 kHz to 15 kHz in increments of 0.5 kHz with a sensitivity of 80 dB re
1uPa. The #D980709 transducer can work at a depth of 6,000 m and in environments
from −10°C to +40°C.

The EdgeTech 8011M model acoustic commander (8011M) is used on board to send
the "ENABLE" command to open the ranging function, the "RANGE" command to
measure the distance between the OBEM and the research vessel, the "DISABLE"
command to close the ranging function, and the "RELEASE1" command to activate
the burn-wire system to release the anchor. The "RELEASE1" command persists for
15 min unless terminated by the "OPTIONAL1" command.




We selected the RF-700A and ST-400A NOVATECH models for the radio and flash
beacons, respectively, for use in the OBEM. The maximum deployment depth for these
models is 7,300 m. The radio beacon is turned ON by sending a VHF signal, and the
flush beacon is turned ON at an atmospheric pressure of less than 1 atm (equal to a
depth of 10 m below the sea surface) in a dark environment. The beacons are also turned
OFF at a depth of 10 m or at an atmospheric pressure of less than 1 atm, respectively.
These two beacons have four independently installed C-type alkaline batteries that
allow for six days of continuous operation at maximum; this power supply differs from
that of the data logger. The two independent power supply layouts allow the beacons to
properly operate even if the power supply for the data logger fails. An on-board radio
scanner detects the signal transmitted from the radio beacon at a distance of 6.4–12.9
km when the OBEM is floating on the surface. These two beacons can assist in locating
the OBEM on the sea surface in both daytime and nighttime.

TL-5930 model lithium batteries manufactured by TADIRAN are used for the OBEM,
with specifications of 3.6 V, 19 Ah, and D-type with characteristics of high energy
density and a low self-discharge rate suitable for long periods of operation. Figure 2
shows a block diagram of the OBEM data logger. The ADC1278EVM model is a 24-
bit A/D converter used for the inputs of the three fluxgate axes, the two tiltmeter axes,
and two pairs of non-polarized electrodes with a sampling rate of 10 SPS. An amplifier
and low-pass filter (Amp & LPF) were designed for the magnetic sensor, leveling sensor,
and electric receiver inputs. The two MPS430F5436A microcontrollers (MCU) process
the timing synchronization of the time base manufactured by SeaSCAN, USA, and the
GPS modules; the digital data is stored to a Secure Digital (SD) memory card with a
standard Secure Digital High Capacity (SDHC), and the user interface communicates
with a PC. The time base module supplies a precise time base signal to the data logger,
whereas the SISMTB Ver 4.1 time base module generates a precise 125-Hz clock that
supports a timing error smaller than $3 \text{ s y}^{-1}$. Even though the time base module supports
a very small timing error of $3 \text{ s y}^{-1}$, the data logger clock is still synchronized with the
GPS on deck for timing corrections after recovering the OBEM. The maximal capacity
of the SD card is 64 GB and can support data storage for more than one year with a
sampling rate of 10 SPS.




Two 17-in glass VITROVEX spheres manufactured by Nautilus Marine Service GmbH,
Germany, are used for the OBEM. These glass spheres contain the fluxgate and tiltmeter
(sensor ball) and the seven channels of the Amp & LPF, data logger, #B980175 ASSY
PCB acoustic controller, and batteries (instrument ball) and can be deployed at a depth
of 6,000 m and support a total buoyancy of 52 kg. The instrument and sensor balls, the
silver chloride electrodes, and the burn-wire system are connected via waterproof
cables. There is a pressure-vacuum valve outside the glass spheres that allows a pumped
vacuum to be preserved at 0.7 atm; self-fusing butyl rubber tape is used to fill the suture
zone between the half glass spheres. In addition, two crossed stainless-steel bands are
used to improve the waterproofing of the glass spheres and cover the orange PE cases.
Four PVC pipes with lengths of 2 m are combined to form the OBEM platform for the
electric receivers, and the silver chloride electrodes are installed at the ends of the pipes.
A 60-kg nonmagnetic anchor is attached to the bottom of the OBEM platform and
catches via a releasing mechanism. The anchor can be released using the burning-wire
system to recover the OBEM. Figure 3 shows a photograph of the OBEM platform.

## 4. Calibrations of the OBEM

It is necessary to calibrate each unit of the OBEM, including the data logger with the
Amp & LPF, fluxgate, tiltmeter, electrodes, ASSY PCB acoustic controller, transducer,
and wiring, before and after assembling the OBEM to improve its performance. We
describe the series of calibration methods used for the OBEM units in the following
section.

**4.1 Calibrations of the background noise of the data logger and the Amp & LPF**

The background noise of the data logger is defined as
$$N_{rms} = \sqrt{\frac{1}{n}(A_1^2 + A_2^2 + \cdots + A_n^2)}, (1)$$
where $n$ is a data point and $A_1$ to $A_n$ indicate the amplitudes of the data points, 1 to n,
individually at short circuit or 0 V. The background noise of the data logger (in "BIT")
is calculated as
$$dB_{rms} = 20\,log_2(N_{rms}), \quad (2)$$





The data logger contains seven input channels called MX, MY, MZ, TX, TY, EX, and
EY. MX, MY, and MZ are used for the magnetic sensor of the fluxgate, TX and TY are
used for the tiltmeter, and EX and EY are used for the electric receivers. The calibration
procedure is described below.

1. Connect MX, MY, MZ, TX, and TY to GND, EX+ with EX−, and EY+ with
   EY−.
2. Start the record mode of the data logger, wait 60 s to acquire data, and then stop
   recording data.
3. Download the data from the data logger and convert it to ASCII format. Then,
   calculate the background noise using Eq. (1) and the background noise in dB
   using Eq. (2).

### 4.2  Calibrations of the sensitivity, linearity error, and dynamic range for the data logger and the Amp & LPF

The input ranges of the voltages for MX, MY, and MZ are ±10 V, for TX and TY are
±5 V, and for EX and EY are ±0.00625 V. The sensitivities are calculated from the
average count of the input voltages, that is, subtract the average count at zero voltage
and then divide by the input voltages:

$$S = Average\left(\frac{Average(C_i) - Average(C_0)}{V_i}\right), \quad (3)$$

where $V_i$ is the input voltage, $C_i$ is the output count saved on the SD card for an input
voltage of $V_i$, and $C_0$ is the output count saved on the SD card for an input voltage of 0
V.
The linearity errors are calculated such that

$$Error = Abs\left[\frac{S_i - S_T}{S_T}\right] \times 100, \quad (4)$$

where $S_i$ is the sensitivity of the input voltage and $S_T$ is the total sensitivity.
The dynamic range is the ratio of the maximum count to the background noise. It is
defined as

$$D = 20\log\left(\frac{S_T \times V\,max}{N_{RMS}}\right), (5)$$

where $S_T$ is the total sensitivity and Vmax is 10 V for MX, MY, and MZ, 5 V for TX




and TY, and 0.00625 V for EX and EY. Its calibration procedure is described below.
1. Connect the MX, MY, and MZ channels of the data logger to the source voltages

generated by the calibrator (FLUKE726) and connect the GND channel of the

data logger to the source common point (COM) of FLUKE726.

2. Set the data logger to the recording mode.
3. Set the FLUKE726 output voltages from 0 V to ±10 V. Increase and decrease

the voltages step by step in 1 V intervals until ±10 V. The measurement time

length for each output voltage is 20 s.

4. Connect the TX and TY channels of the data logger to the source voltages

generated by FLUKE726 and connect the GND channel of the data logger to

COM of FLUKE726.

5. Set the FLUKE726 output voltages from 0 V to ±5 V. Increase and decrease the

voltages step by step in 1 V intervals until ±5 V. The measurement time length

for each output voltage is 20 s.

6. Connect the EX+ and EY+ channels of the data logger to the source voltages

generated by FLUKE726, and connect the EX− and EY− channels of the data

logger to COM of FLUKE726.

7. Set the FLUKE726 output voltages from 0 V to ±6 mV. Increase and decrease

the voltages step by step in 1-mV intervals until ±6 mV. The measurement time

length for each output voltage is 20 s.

8. Finally, switch off the recording mode of the data logger, download the data,

and convert it to ASCII format for analysis. Calculate the sensitivity, linearity

error, and dynamic range using Eqs. (3), (4), and (5), respectively.


Tables 1–3 show the results for the background noise, sensitivity, linearity error, and
dynamic range of the calibrations of the magnetic (MX, MY, and MZ), electric (EX
and EY), and tiltmeter (TX and TY) channels of the OBEM01 data logger and the Amp
& LPF. Figure 4 shows an example calibration of the magnetic channels checking the
sensitivity, linearity, and error. The average sensitivity is 655,968.5 counts/V with a
maximum error smaller than 1.35%. Figure 5 shows an example calibration of the
electric channels checking the sensitivity, linearity, and error. The average sensitivity
is 135,856,047.8 counts/V with a maximum error smaller than 0.8%. Figure 6 shows
an example calibration of the tiltmeter channels checking the sensitivity, linearity, and



error. The average sensitivity is 1,677,710.6 counts/V with a maximum error smaller
than 0.25%.

**4.3 Evaluation of the current consumption**
The power supplies of the OBEM consist of two 7.2-V battery packs in a series
connection with two 3.6-V lithium batteries. One battery pack is for the data logger and
converts to ±5 VDC and +3.3 VDC. The other pack is for the sensors and converts to
±5 VDC and +12.0 VDC. Two +7.4-VDC output current batteries were measured for
their current consumption measurement using two ammeters connecting the two +7.4-
V battery packs. Table 4 shows the current consumption of the OBEM system. The
maximum current consumptions of the data logger and sensors are 32 mA and 105 mA,
respectively. The total power consumption is less than 1 W, which corresponds to
expectations.

**4.4 Evaluation of the electrodes**
Two pairs of silver chloride electrodes are used for the OBEM. We first put a pair of
electrodes separated by a fixed distance within a tank filled with seawater to check the
status of the electrodes. Second, we measured the electrical potential and impedance of
the electrodes using a digital volt-ohm-milliammeter (VOM) (Fig. 7). Third, we sent a
sweep sine signal to check the frequency responses of the electrodes, as shown in Fig.
8. Fourth, we input a DC voltage to check the electrode-induced voltages, as shown in
Fig. 9. Table 5 shows the self-potential, impedance, and induced voltages for each pair
of electrodes. The ranges of the self-potential and impedance are 0.26–3.63 mV and
243–370 Ω, respectively. The electrical potential shows that 81–167 mV was
transmitted from the 5 VDC of the two copper electrodes.

**4.5 Evaluation of the fluxgate**
The fluxgate is mounted in the sensor ball of the OBEM. Therefore, we could only
calculate the total magnetic field (TMF) (Eq. (6)) measured from the three components
of the fluxgate. We then compared the difference between the TMF of the OBEM and
geomagnetic data of the geophysical database management system from the Central
Weather Bureau. The TMF is calculated by


$$M_T = \sqrt{(M_X^2 + M_Y^2 + M_Z^2)}, \quad (6)$$

where $M_X$, $M_Y$, and $M_Z$ are the components of the north–south, east–west, and vertical
magnetic fields, respectively.

**4.6 Evaluation of the acoustic transceiver and its transducer**
We selected the large-scale Breeze Canal in New Taipei City for testing because it has
few obstacles and is suitable for evaluating the functions of the 8011M. The Breeze
Canal has a length of approximately 800 m and is located in a straight river with a depth
of 2–5 m. The distance between the transducer and the acoustic transceiver was
approximately 630 m, and the layout for the field test is shown in Fig. 10. The testing
procedure for the transducers is described below. The results are listed in Table 6.
1. Connect the tested transducer and acoustic transceiver via an underwater cable,

and place the tested transducer and transceiver at an underwater depth of 1 m.

2. Record the serial numbers of the transducers in a notebook.
3. Send the "ENABLE" command via the 8011M, and then count the response

beeps.

4. Send the "RANGE" command via the 8011M five times, and record the distance

of each ranging.

5. Send the "DISABLE" command via the 8011M, and then count the response

beeps.

6. Replace the transducer, and return to step 2 to repeat the evaluation.

We then checked the acoustic transceivers after all of the transducers were successfully
checked; the testing procedure for the acoustic controller is described below. The results
are listed in Table 7.
1. Change the acoustic controller, and record its serial number in a notebook.
2. Send the "ENABLE" command via the 8011M, and then count the response

beeps.

3. Send the "RANGE" command via the 8011M five times, and record the distance

of each ranging.

4. Send the "RELEASE1" command via the 8011M, and then count the response

beeps. Check the voltage between Pin1 and Pin2 of JP2 using a VOM. It should


be greater than 12.0 VDC.
5. Send the "OPTION1" command via the 8011M, and then count the response

beeps. Check the voltage between Pin1 and Pin2 of JP2 using a VOM. It should

be 0 VDC.

6. Send the "RELEASE2" command via the 8011M, and then count the response

beeps. Check the voltage between Pin3 and Pin4 of JP2 using a VOM. It should

be greater than 12.0 VDC.

7. Send the "OPTION1" command via the 8011M, and then count the response

beeps. Check the voltage between Pin3 and Pin4 of JP2 using a VOM. It should

be 0 VDC.

8. Send the "DISABLE" command via the 8011M, and then count the response

beeps.

9. Send the "RANGE" command via the 8011M; there should be no response from

the transceiver.

10. Return to step 1 to repeat the evaluation.
A mercury switch is mounted on the transceiver which when turned off responds with
15 beeps and when turned on responds with seven beeps.

**5.    The preliminary result of the OBEM offshore Taiwan**
We deployed six broadband BBYBs and one OBEM near a small submarine volcano
area in the OT offshore NE Taiwan (Fig. 11) on 03/26/2018 for a submarine observation
to evaluate all the OBEM units. All the equipment was successfully recovered after one
month of deployment. Figure 12 shows the time series data of OBEM01. The TMF
calculated from the three components of the magnetic field varied in the range of
44,100–4,4150 nT, which corresponded to the geomagnetic field measured by proton
magnetometers in Taiwan. The two horizontal magnetic fields contained significant
daily variations. Furthermore, the vibrations of the inclinations were significantly
affected by two earthquakes on 04/27/2018 (at 12:41 UTC and 12:47 UTC) consistent
with seismic signals of the BBYBs (Fig. 13). The average magnetic fields of HX, HY,
HZ, and TMF 2 s prior to the earthquakes (12:41 UTC) were 12,900 nT, 34,300 nT,
24,600 nT, and 44137 nT, respectively, the average potential fields of EX and EY were
−0.79 mV and −0.149 mV, respectively, and the inclinations of TX and TY were −2.65°



and 1.21°, respectively. These were the averages of the background without earthquakes.

We subtracted the background averages of the magnetic fields and the inclinations to
compare the differential during the 12:41 UTC event as shown in Fig. 14. The peak
ground motion velocity (PGV) was 2.63 cm s$^{-1}$ on the SH1 corresponding to
inclinations of 0.4° and 0.6° for TX and TY with a 100 nT disturbance of HY. There
was an insignificant amount of variation in the electric fields. The result shows that the
earthquake significantly affected the HY component.

**6.  Conclusions**
A long-period OBEM acquisition platform to measure magnetic and electrical fields on
the seafloor was successfully constructed and evaluated by the OBS R&D team for
deployment offshore Taiwan. The power consumption of the OBEM is less than 1 W,
which means that the lifetime could be extended up to 300 days with the installation of
108 lithium batteries. We deployed and recovered the OBEM at an underwater depth of
1,400 m to acquire the first marine magnetotelluric data offshore NE Taiwan.

Six broadband BBYBs and one OBEM were deployed near a small submarine volcano
area offshore NE Taiwan. The TMF calculated from the three magnetic field
components varied in the range of 44,100–4,4150 nT, which corresponded to the proton
magnetometer measurements of the geomagnetic field in Taiwan. The two horizontal
magnetic fields displayed significant daily variations, and the vibrations of the
inclinations were significantly affected by the two earthquakes that occurred during the
observations. There was an insignificant amount of variation in the electric fields.

Localized micro-earthquakes affected the disturbances of magnetic field and
inclinations in this study. Therefore, to improve the efficacy of marine geophysical
explorations, a platform for multiple underwater measurements is required including an
ocean bottom flow meter, thermometer, and absolute pressure gage. We will focus on
such developments in the future.

**Acknowledgments**
We greatly appreciate the crews of R/V OR2 for the field experiments. The authors



acknowledge the financial support from the Ministry of Science and Technology of
Taiwan under grant numbers of 105-2116-M-019-001, 106-2116-M-001-008, 106-
2116-M-019-003, and 107-2116-M-019-006. We also thank four years of the Taiwan-
German cooperative projects on gas hydrate of NEPII for supporting the funds of the
instrument deployment of the OBEMs. We would like to thank the TEC Data Center
for proving graphical services.

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




**TABLE AND FIGURE CAPTIONS**

Table 1. The OBEM01 data logger calibration of the magnetic channels with Amp &
LPF: the background noise, sensitivity, linearity error, and dynamic range.

Table 2. The OBEM01 data logger calibration of the electric channels with Amp & LPF:
the background noise, sensitivity, linearity error, and dynamic range.

Table 3. The OBEM01 data logger calibration of the tiltmeter channels with Amp &
LPF: the background noise, sensitivity, linearity error, and dynamic range.

Table 4. The total current consumption of the OBEMs.

Table 5. The self-potential, impedance, and induced voltage results for each pair of
silver chloride electrodes.

Table 6. Example results for the functional test of the acoustic transducer.

Table 7. Example results for the functional test of the acoustic controller.

Figure 1. A block diagram of the OBEM. The inputs of the two electric fields, two
inclinations, and three magnetic fields pass through the Amp & LPF in the data logger,
which contains a 64-GB SD card. The SeaSCAN time base module is integrated into
the data logger and has a timing error smaller than 3 s $y^{-1}$. The EdgeTech acoustic
transceiver and transducer are used for the positioning and releasing of the anchor. The
radio and flash beacons are used to locate the OBEM at the sea surface during recovery
operations.

Figure 2. A block diagram of the OBEM data logger. The ADS1278EVM is a 24-bit
A/D with eight inputs used for converting analog signals via the amplifier and low-pass
filter (Amp & LPF) to digital data. The Amp & LPF adjusts the output voltages of the
sensors of the fluxgate, tiltmeter, and electric receivers to suitable A/D input levels. The
two MCUs of the MPS430F5436A process the timing synchronization by the



SeaSCAN of time base and GPS modules, the digital data storage to the SD card with
a standard SDHC, and the user interface communication with a PC.

Figure 3. A photograph of the OBEM01 and its specific modules.

Figure 4. Calibration results for the magnetic channels of the OBEM01. The average
sensitivity is 655,968.5 counts/V, and the maximum error is <1.35%.

Figure 5. Calibration results for the electric channels of the OBEM01. The average
sensitivity is 1,358,568,047.8 counts/V, and the maximum error is <0.8%.

Figure 6. Calibration results for the inclination channels of the OBEM01. The average
sensitivity is 1,677,710.6 counts/V, and the maximum error is <0.25%.

Figure 7. The layout for the evaluation of the electric receivers. Two copper electrodes
are used to vary the input signals. A pair of silver chloride electrodes are placed at the
corner of a tank with an area of 68 cm × 49 cm filled with 15 cm of seawater. A VOM
is used to measure the self-potential and impedance of the electrodes.

Figure 8. The responses of the electrodes with varying frequencies. The response curves
of $V_o/V_i$ are proportional to the frequency on a log scale.

Figure 9. The responses of the electrodes with varying voltages. The input was ranged
from 500 mVDC to 2,500 mVDC to check the induced voltage; the induced voltages
are proportional to the input voltages.

Figure 10. A map of the field test to evaluate the acoustic transducer, acoustic controller,
and 8011M.

Figure 11. A location map showing the BBYBs and OBEM with triangle and diamond
symbols, respectively.





Figure 12. The OBEM01 time series data. The panels from top to bottom in the figure
show the four magnetic fields: TMF, HX, HY, and HZ, the two electric fields: EX and
EY, and the two inclinations: TX and TY.

Figure 13. Comparison of the OBEM01 and 1802OBS time series data during the two
earthquakes. The two earthquakes affected the inclinations. The first and secondary
earthquakes occurred at 12:41 UTC and 12:47 UTC, respectively, on 04/27/2018.

Figure 14. The variations in PGV, TMF, HY, TX, and TY during the first earthquake.
The PGV of 2.63 cm/s affected the inclinations by 0.601° and 0.404° for TX and TY,
respectively, and the HY magnetic field had a peak of 100 nT.





**TABLES AND FIGURES**

## Table 1

| input (V) | output(MX) | remove offset(MX) | output(MY) | remove offset(MY) | output(MZ) | remove offset(MZ) | input | sensitivity(MX) | error%(MX) | sensitivity(MY) | error%(MY) | sensitivity(MZ) | error%(MZ) |
|---|---|---|---|---|---|---|---|---|---|---|---|---|---|
| -10.0 | -6471112.3 | -6474180.3 | -6472546.0 | -6471775.5 | -6476434.0 | -6472369.2 | -10.0 | 647418.03 | -1.322 | 647177.55 | -1.342 | 647236.92 | -1.311 |
| -9.0 | -5869208.1 | -5872276.1 | -5871019.8 | -5870249.3 | -5874491.2 | -5870426.4 | -9.0 | 652475.12 | -0.551 | 652249.92 | -0.568 | 652269.60 | -0.543 |
| -8.0 | -5249375.0 | -5252443.1 | -5251749.3 | -5250978.8 | -5254300.4 | -5250235.6 | -8.0 | 656555.38 | 0.070 | 656372.35 | 0.060 | 656279.45 | 0.068 |
| -7.0 | -4600873.9 | -4603941.9 | -4603731.6 | -4602961.1 | -4605554.6 | -4601489.8 | -7.0 | 657705.99 | 0.246 | 657565.87 | 0.242 | 657355.68 | 0.232 |
| -6.0 | -3943651.8 | -3946719.8 | -3946684.9 | -3945914.4 | -3948745.5 | -3944680.7 | -6.0 | 657786.64 | 0.258 | 657652.39 | 0.255 | 657446.78 | 0.246 |
| -5.0 | -3285800.0 | -3288868.0 | -3289015.6 | -3288245.1 | -3291209.9 | -3287145.1 | -5.0 | 657773.61 | 0.256 | 657649.02 | 0.255 | 657429.02 | 0.243 |
| -4.0 | -2628274.7 | -2631342.8 | -2631609.2 | -2630838.7 | -2634025.5 | -2629960.7 | -4.0 | 657835.69 | 0.266 | 657709.67 | 0.264 | 657490.16 | 0.253 |
| -3.0 | -1970402.2 | -1973470.2 | -1973864.9 | -1973094.4 | -1976483.5 | -1972418.7 | -3.0 | 657823.41 | 0.264 | 657698.14 | 0.262 | 657472.91 | 0.250 |
| -2.0 | -1312631.3 | -1315699.4 | -1316216.6 | -1315446.1 | -1319057.9 | -1314993.0 | -2.0 | 657849.68 | 0.268 | 657723.04 | 0.266 | 657496.52 | 0.254 |
| -1.0 | -654832.3 | -657900.3 | -658536.9 | -657766.4 | -661600.3 | -657535.4 | -1.0 | 657900.33 | 0.275 | 657766.39 | 0.273 | 657535.43 | 0.260 |
| 0.0 | 3068.0 | 0.0 | -770.5 | 0.0 | -4064.8 | 0.0 | 1.0 | 657880.86 | 0.272 | 657759.33 | 0.271 | 657535.67 | 0.260 |
| 0.0 | 3018.0 | 0.0 | -810.1 | 0.0 | -4118.9 | 0.0 | 2.0 | 657859.96 | 0.269 | 657741.27 | 0.269 | 657504.87 | 0.255 |
| 1.0 | 660948.9 | 657930.8 | 656988.8 | 657798.9 | 653470.8 | 657589.8 | 3.0 | 657837.12 | 0.266 | 657727.37 | 0.267 | 657491.44 | 0.253 |
| 2.0 | 1318787.9 | 1315769.9 | 1314712.0 | 1315522.1 | 1310944.9 | 1315063.8 | 4.0 | 657859.68 | 0.269 | 657747.51 | 0.270 | 657499.55 | 0.254 |
| 3.0 | 1976579.4 | 1973561.4 | 1972411.6 | 1973221.7 | 1968409.5 | 1972528.4 | 5.0 | 657811.97 | 0.262 | 657692.96 | 0.261 | 657443.49 | 0.245 |
| 4.0 | 2634506.8 | 2631488.7 | 2630219.5 | 2631029.6 | 2625933.4 | 2630052.3 | 6.0 | 657832.48 | 0.265 | 657705.29 | 0.263 | 657459.85 | 0.248 |
| 5.0 | 3292127.9 | 3289109.8 | 3287694.3 | 3288504.4 | 3283152.6 | 3287271.6 | 7.0 | 657688.48 | 0.243 | 657584.21 | 0.245 | 657397.26 | 0.238 |
| 6.0 | 3950062.9 | 3947044.9 | 3945461.2 | 3946271.3 | 3940694.3 | 3944813.2 | 8.0 | 656372.97 | 0.043 | 656448.69 | 0.072 | 656415.89 | 0.089 |
| 7.0 | 4606887.4 | 4603869.3 | 4602318.9 | 4603129.0 | 4597716.0 | 4601834.9 | 9.0 | 652314.56 | -0.576 | 652334.15 | -0.556 | 652469.16 | -0.513 |
| 8.0 | 5254051.8 | 5251033.8 | 5250819.0 | 5251629.1 | 5247262.3 | 5251381.2 | 10.0 | 647283.74 | -1.343 | 647270.99 | -1.327 | 647438.99 | -1.280 |
| 9.0 | 5873899.1 | 5870881.0 | 5870236.8 | 5871046.9 | 5868157.6 | 5872276.5 | Average | 656093.29 | | 655978.81 | | 655833.43 | |
| 10.0 | 6475905.4 | 6472887.3 | 6471939.4 | 6472749.5 | 6470325.1 | 6474444.0 | **Average sensitivity 655968.51** | | | | | | |



## Table 2

| Input (V) | Output (EX) | remove offset (EX) | Output (EY) | remove offset (EY) | input (V) | sensitivity (EX) | error%(EX) | sensitivity (EY) | error% (EY) |
|---|---|---|---|---|---|---|---|---|---|
| -0.0060 | -7973544.4 | -8134716.5 | -8135152.5 | -8127780.7 | -0.0060 | 1355786076.8 | -0.186 | 1354630119.94 | -0.413 |
| -0.0050 | -6611209.2 | -6772381.3 | -6778699.0 | -6771327.2 | -0.0050 | 1354476260.4 | -0.283 | 1354265449.87 | -0.439 |
| -0.0040 | -5257318.1 | -5418490.2 | -5459462.4 | -5452090.6 | -0.0040 | 1354622556.8 | -0.272 | 1363022659.48 | 0.204 |
| -0.0030 | -3909730.8 | -4070902.9 | -4084816.3 | -4077444.6 | -0.0030 | 1356967645.6 | -0.099 | 1359148194.55 | -0.081 |
| -0.0020 | -2558460.2 | -2719632.3 | -2729127.4 | -2721755.7 | -0.0020 | 1359816155.1 | 0.110 | 1360877857.04 | 0.047 |
| -0.0010 | -1207366.1 | -1368538.2 | -1376888.8 | -1369517.1 | -0.0010 | 1368538175.4 | 0.753 | 1369517055.70 | 0.682 |
| 0.0000 | 161172.1 | 0.0 | -7371.7 | 0.0 | 0.0010 | 1368374397.6 | 0.740 | 1368868765.94 | 0.634 |
| 0.0000 | 95647.7 | 0.0 | -86742.8 | 0.0 | 0.0020 | 1359089788.7 | 0.057 | 1361166374.27 | 0.068 |
| 0.0010 | 1464022.1 | 1368374.4 | 1282126.0 | 1368868.8 | 0.0030 | 1357537440.6 | -0.057 | 1359385875.47 | -0.063 |
| 0.0020 | 2813827.3 | 2718179.6 | 2635589.9 | 2722332.7 | 0.0040 | 1354770300.9 | -0.261 | 1351072742.78 | -0.674 |
| 0.0030 | 4168260.1 | 4072612.3 | 3991414.8 | 4078157.6 | 0.0050 | 1354197292.1 | -0.303 | 1349041143.00 | -0.824 |
| 0.0040 | 5514728.9 | 5419081.2 | 5317548.2 | 5404291.0 | 0.0060 | 1355623271.0 | -0.198 | 1354837547.58 | -0.397 |
| 0.0050 | 6866634.2 | 6770986.5 | 6658462.9 | 6745205.7 | Average | 1358316613.4 | | 1358819482.1 | |
| 0.0060 | 8229387.4 | 8133739.6 | 8042282.5 | 8129025.3 | **Average sensitivity 1358568047.8** | | | | |



Table 3

| input(V) | output(TX) | remove offset(TX) | output(TY) | remove offset(TY) | input | sensitivity(TX) | error%(TX) | sensitivity(TY) | error%(TY) |
|---|---|---|---|---|---|---|---|---|---|
| -5.00 | -8387520.5 | -8386705.4 | -8387650.3 | -8386680.2 | -5.0 | 1677341.1 | -0.016 | 1677336.04 | -0.029 |
| -4.00 | -6712516.4 | -6711701.2 | -6712951.5 | -6711981.4 | -4.0 | 1677925.3 | 0.019 | 1677995.36 | 0.011 |
| -3.00 | -5034477.1 | -5033661.9 | -5034843.7 | -5033873.6 | -3.0 | 1677887.3 | 0.017 | 1677957.86 | 0.008 |
| -2.00 | -3356824.4 | -3356009.2 | -3357098.5 | -3356128.4 | -2.0 | 1678004.6 | 0.024 | 1678064.22 | 0.015 |
| -1.00 | -1678971.4 | -1678156.3 | -1679215.6 | -1678245.5 | -1.0 | 1678156.3 | 0.033 | 1678245.46 | 0.026 |
| 0.00 | -815.2 | 0.0 | -970.1 | 0.0 | 1.0 | 1678289.0 | 0.041 | 1678377.40 | 0.033 |
| 0.00 | -902.3 | 0.0 | -1060.3 | 0.0 | 2.0 | 1678208.8 | 0.036 | 1678269.99 | 0.027 |
| 1.00 | 1677386.7 | 1678289.0 | 1677317.1 | 1678377.4 | 3.0 | 1678386.2 | 0.047 | 1678923.87 | 0.066 |
| 2.00 | 3355515.3 | 3356417.6 | 3355479.6 | 3356540.0 | 4.0 | 1678457.5 | 0.051 | 1679001.78 | 0.071 |
| 3.00 | 5034256.3 | 5035158.6 | 5035711.3 | 5036771.6 | 5.0 | 1673403.6 | -0.250 | 1673981.58 | -0.228 |
| 4.00 | 6712927.8 | 6713830.2 | 6714946.8 | 6716007.1 | Average | 1677606.0 | | 1677815.36 | |
| 5.00 | 8366115.7 | 8367018.0 | 8368847.6 | 8369907.9 | **Average sensitivity** | | | **1677710.6** | |

Table 4

| Logger S/N | Turn-on Mode (mA) | | | Recording Mode (mA) | | |
|---|---|---|---|---|---|---|
| | 7.2V for Data logger | 7.2V for Sensors | Power consumption | 7.2V for Data logger | 7.2V for Sensors | Power consumption |
| OBEM01 | 32 | 104 | 0.98 | 31 | 105 | 0.98 |
| OBEM02 | 30 | 94 | 0.89 | 29 | 97 | 0.91 |
| OBEM03 | 29 | 103 | 0.95 | 29 | 104 | 0.96 |

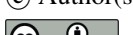


Table 5

|  | Electrical potential | Impedance | Input DC5V, induce voltage |
|---|---|---|---|
| OBEM01(EX) | 0.56 mV | 245 Ω | 164 mV |
| OBEM01(EY) | 0.26 mV | 272 Ω | 167 mV |
| OBEM02(EX) | 3.63 mV | 243 Ω | 81 mV |
| OBEM02(EY) | 1.93 mV | 370 Ω | 95 mV |
| OBEM03(EX) | 2.38 mV | 267 Ω | 83 mV |
| OBEM03(EY) | 2.1 mV | 331 Ω | 83 mV |

Table 6

| Transducer S/N | Enable Beep (Times) | Disable Beep (Times) | 1st Ranging Distance show on 8011M (m) | 2nd Ranging Distance show on 8011M (m) | 3rd Ranging Distance show on 8011M (m) | 4th Ranging Distance show on 8011M (m) | 5th Ranging Distance show on 8011M (m) | Judgment |
|---|---|---|---|---|---|---|---|---|
| 35427 | 15 | 15 | 629 | 628 | 630 | 627 | 628 | Good |
| 35428 | 15 | 15 | 629 | 627 | 629 | 630 | 629 | Good |
| 35429 | 15 | 15 | 630 | 630 | 630 | 629 | 629 | Good |

Table 7

| S/N | Enable Beep (Times) | 1st Ranging Distance show on 8011M (m) | 2nd Ranging Distance show on 8011M (m) | 3rd Ranging Distance show on 8011M (m) | 4th Ranging Distance show on 8011M (m) | 5th Ranging Distance show on 8011M (m) | RELEASE1 Beep Times/Volt | OPTION1 Beep (Times) | RELEASE2 Beep Times/Volt | OPTION1 Beep (Times) | DISABLE Beep (Times) |
|---|---|---|---|---|---|---|---|---|---|---|---|
| 50854 | 15 | 628 | 629 | 630 | 630 | 630 | 15/ 12.77V | 15 | 15/ 12.77V | 15 | 15 |
| 50784 | 7 | 629 | 630 | 630 | 630 | 630 | 7/ 12.77V | 7 | 7/ 12.77V | 7 | 7 |
| 50783 | 15 | 628 | 628 | 628 | 629 | 631 | 15/ 12.77V | 15 | 15/ 12.77V | 15 | 15 |



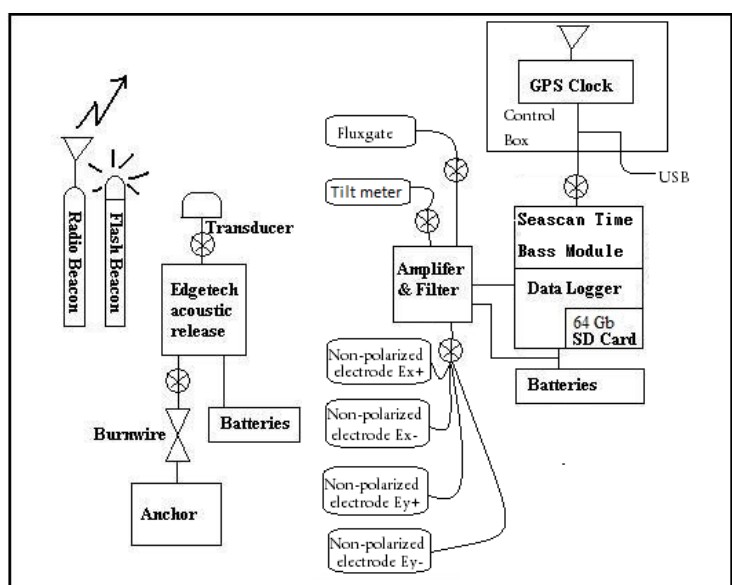

Figure 1

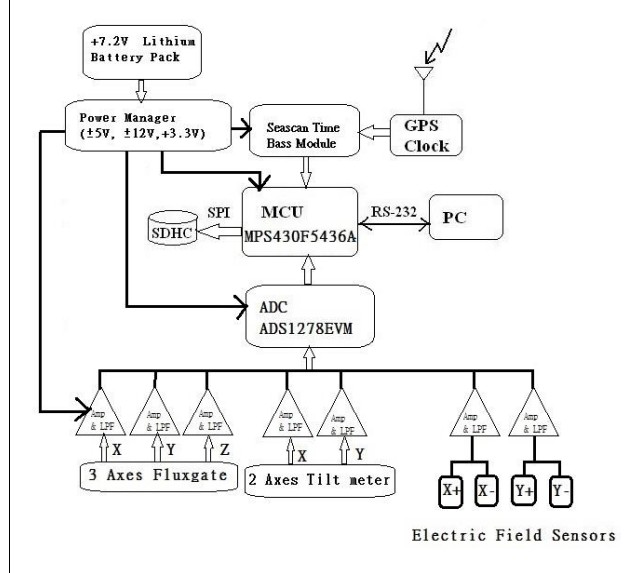

Figure 2

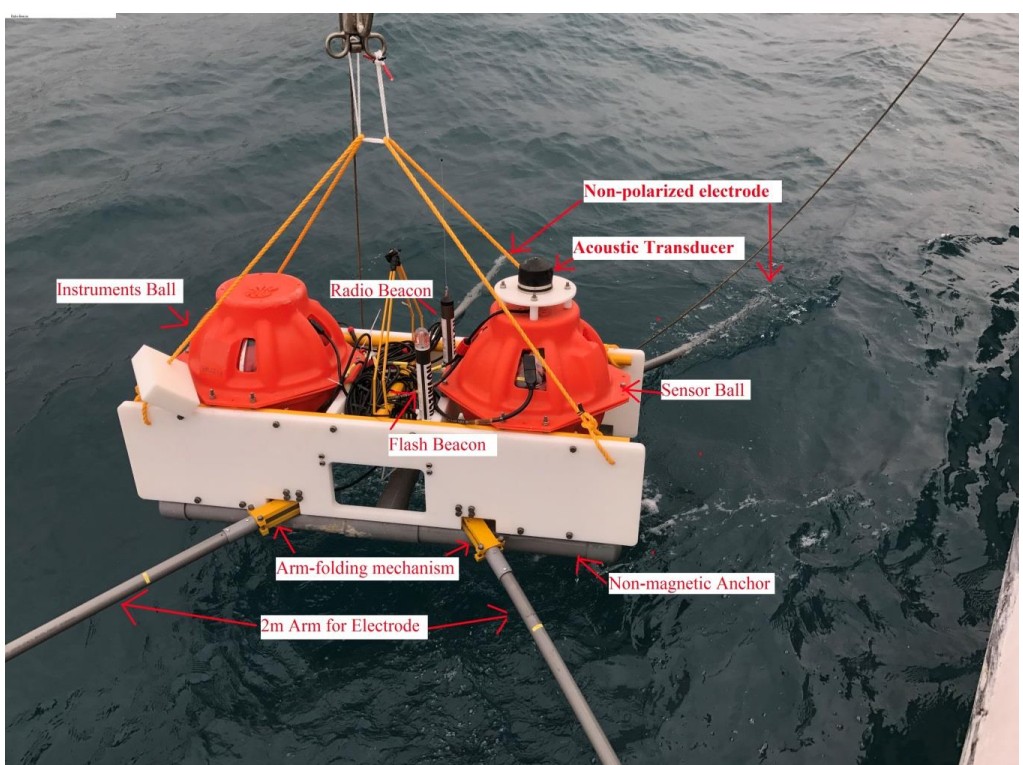

Figure 3

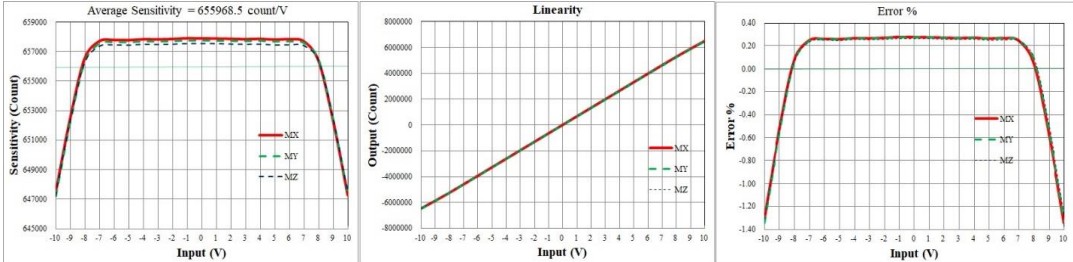

Figure 4





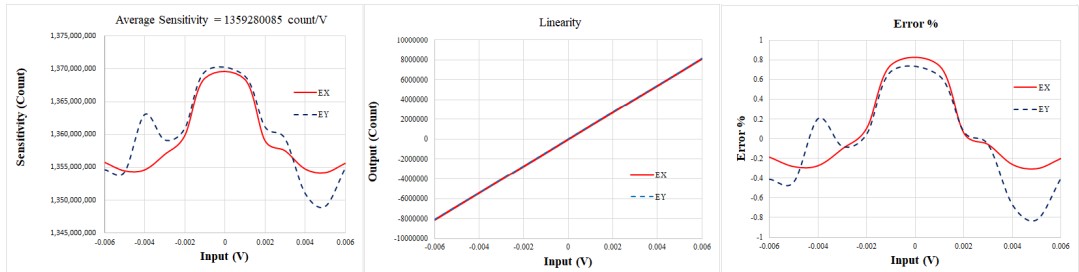

Figure 5

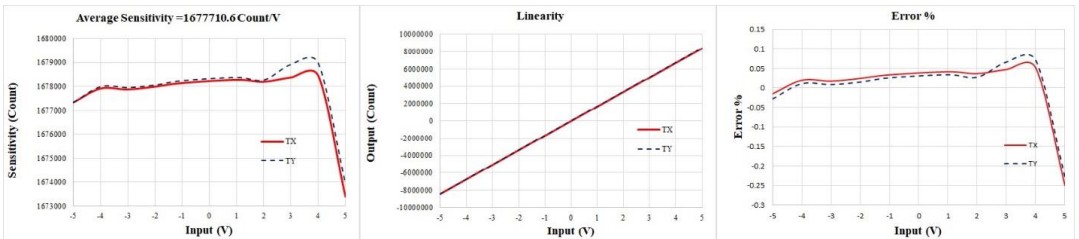

Figure 6

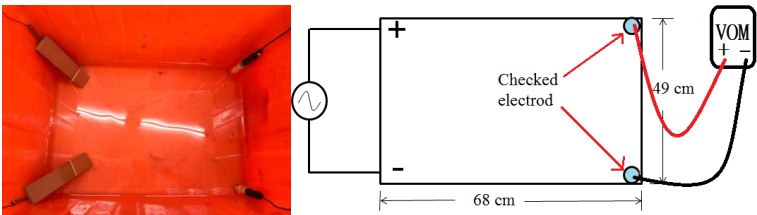

Figure 7




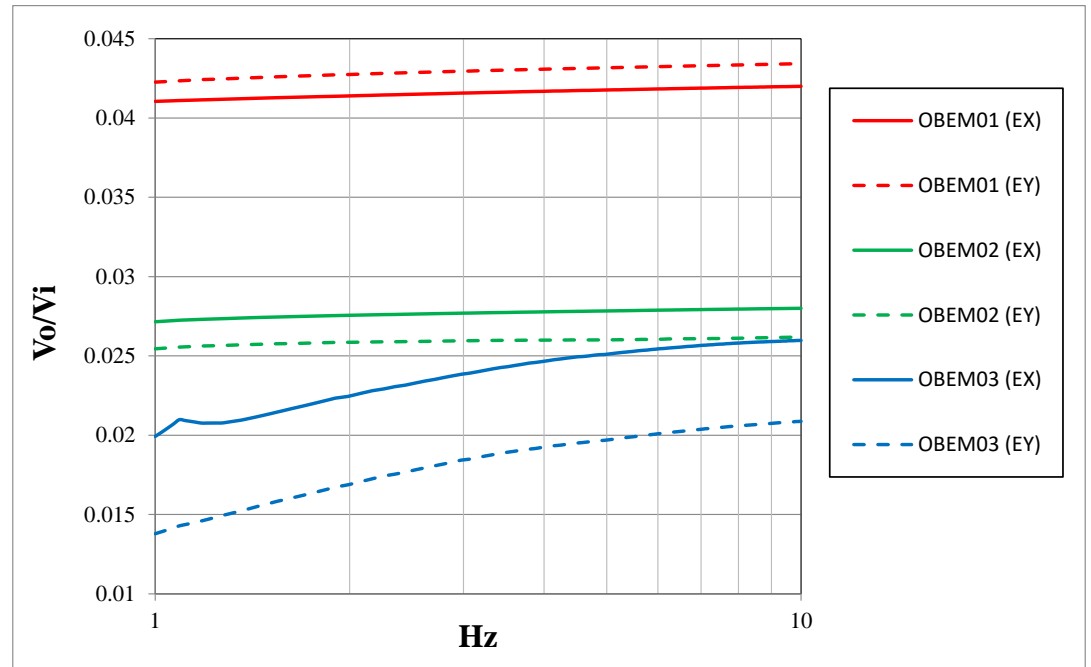

Figure 8

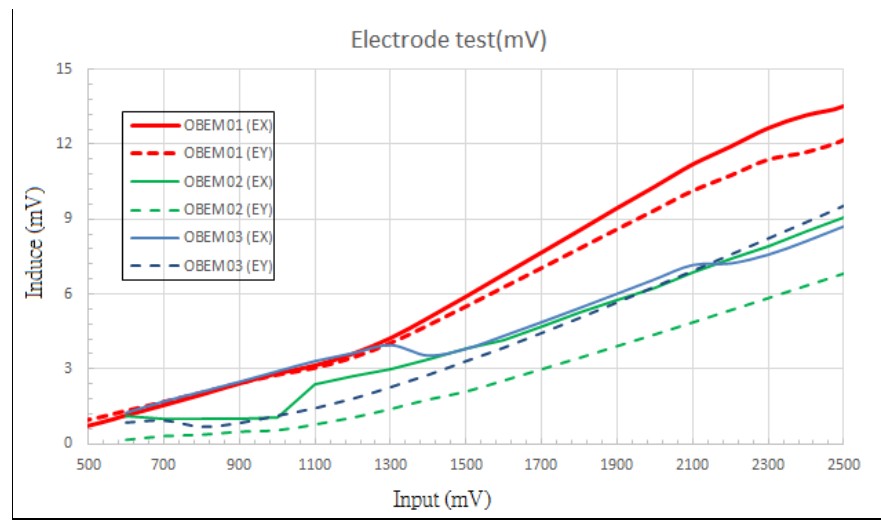

Figure 9





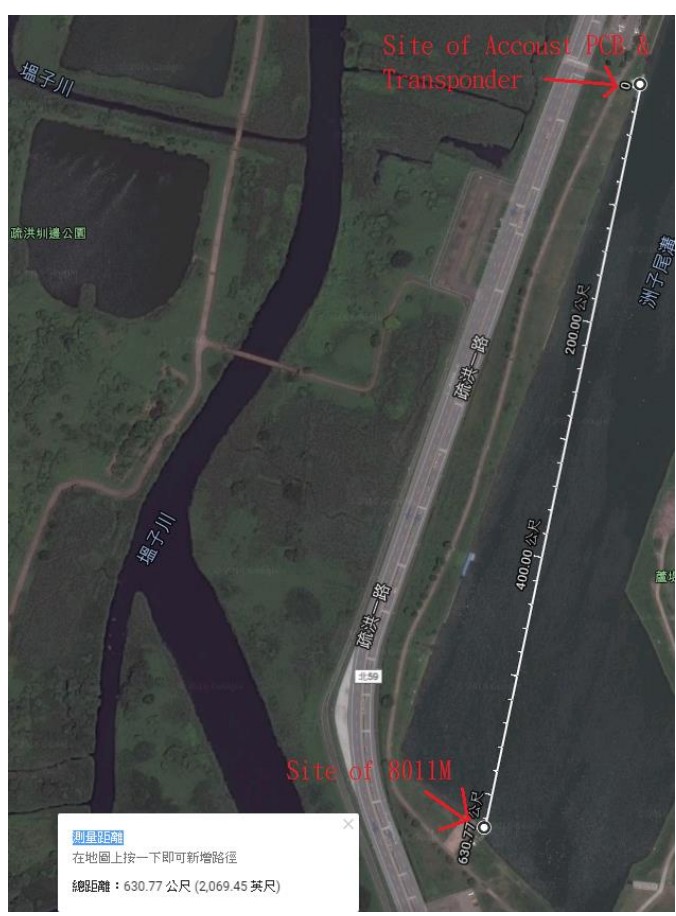

Figure 10





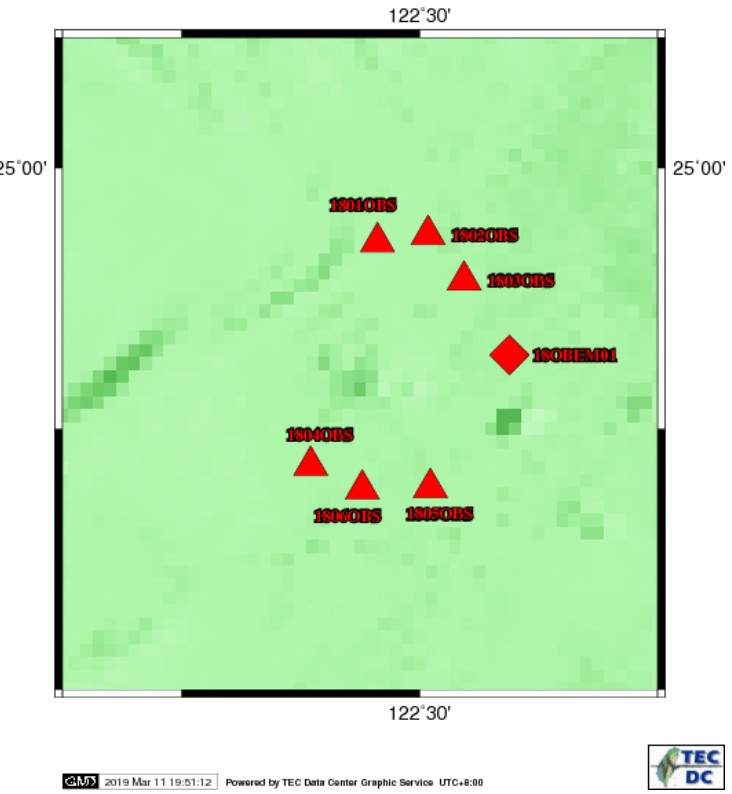

Figure 11




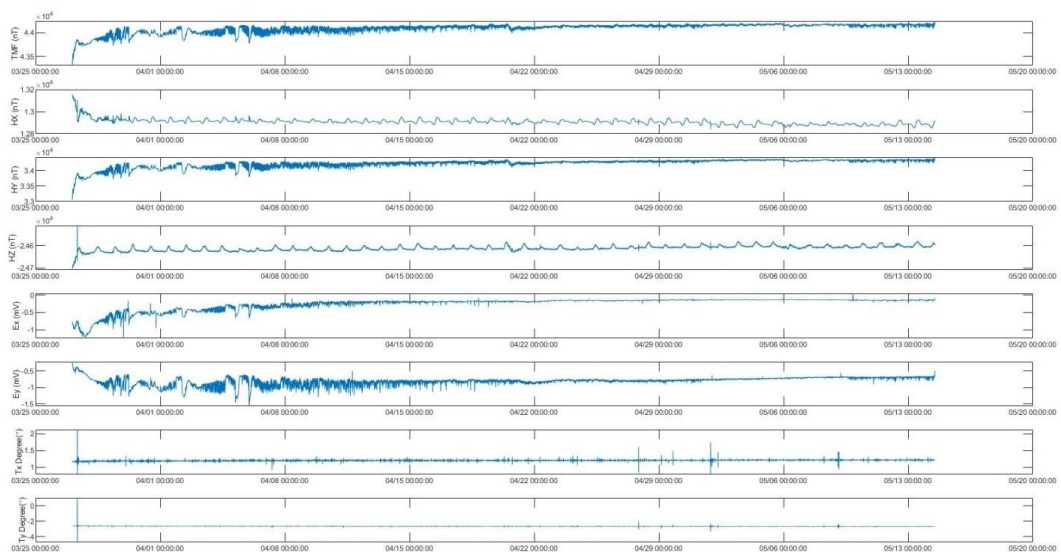

Figure 12

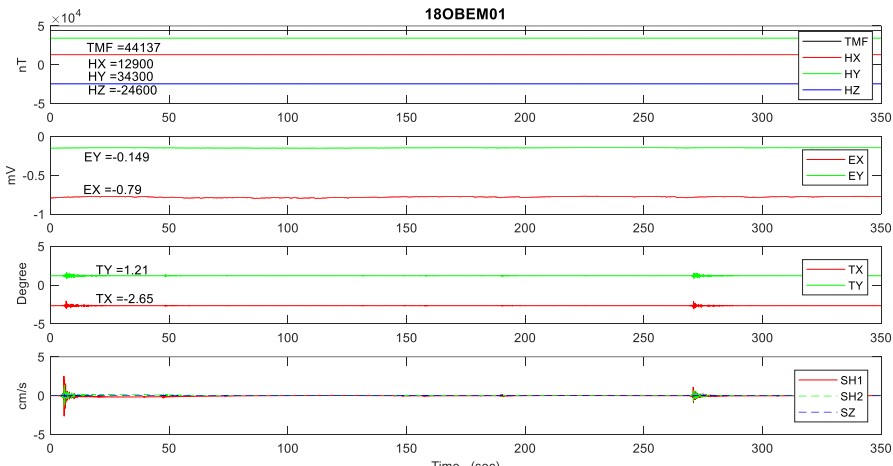

Figure 13





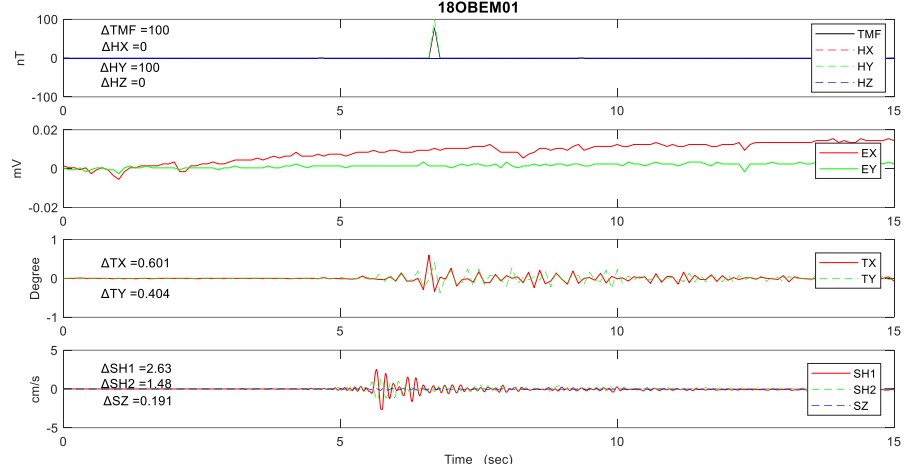

Figure 14