# Peer review of "Evaluations of an ocean bottom electro-magnetometer and preliminary results offshore NE Taiwan"

_Geoscientific Instrumentation, Methods and Data Systems, 2019_

## Referee Comment (RC1) · 24 Jul 2019

Review of GI-2019-13 "Evaluations of an ocean bottom electro-magnetometer and preliminary results offshore NE Taiwan" by Ching-Ren Lin et al.

In this paper, you proposed an ocean bottom electromagnetic receiver (OBEM) for marine MT data acquisition. I think your achievement in this paper is very interesting. However, basic concept of your OBEM was almost based on existing OBEM (ksaya,2009). Hence, I suggest you enhance the difference and advantages of your instruments. This paper also discussed the field operation and data, but there is a lack of necessary data and discussion. Therefore, my decision of this paper is "major revision". My questions

and other suggestions are as follows.

1. I suggest add important result of marine MT data acquisition, such as Coherence between horizontal magnetic components and the electric field component, and the Apparent resistivity and impedance phase.

2. I suggest add compare with the OBE and OBEM from (kasay, 2009),Qmax3 from Quasar. Add a table present the key specification such as Nosie level, dynamic, power consumption, time drift error size, and cost.

3. The details of electrode, fluxgate sensor and amplifier of your OBEM are not described.

4. Figure and tables should be clear and simply. Table1-3 can be replaced with figure. the tables are too large to reduce paper space; figure 1and figure 2 should be redraw to clear; figure4-6, the font size too small to read. Background in Figure 11 is too vague to read.

5. The ADS1278 is suitable for audio frequency signal measurement, but the samples is 10Hz, the ADS 1278 is not the best choice. Why choose ADS1278?

6. The abstract should enhance the innovation design of the OBEM compare the existing OBEM.

7. Line 37-line 40 and L45-L49 introduce CSEM method. But your OBEM only used to MT data acquisition. I suggest to delete this sentences.

8. The OBEM integrated the beacon (RF-700A and ST400A) which are expensive. Why not developed a beacon module just like MicrOBS from Sercel?

9. The fluxgate sensor is installed in glass sphere, the distance between data logger and sensor may be very short, how to reduce the disturbance (EM noise) from data logger while writing data to SD?

10. In 4.1, the description of noise and linearity is too simple to give details. And the

test of release is too detailed because the release is from EdgeTech not developed by author.

11. Reference 2 is the same as reference 3?

Kai Chen Associate professor China University of Geosciences, Beijing, China ck@cugb.edu.cn

---

## Author Comment (AC1) · 1 Aug 2019

Review of GI-2019-13 "Evaluations of an ocean bottom electro-magnetometer and preliminary results offshore NE Taiwan" by Ching-Ren Lin et al. In this paper, you proposed an ocean bottom electromagnetic receiver (OBEM) for marine MT data acquisition. I think your achievement in this paper is very interesting. However, basic concept of your OBEM was almost based on existing OBEM (ksaya,2009). Hence, I suggest you enhance the difference and advantages of your instruments. This paper also discussed the field operation and data, but there is a lack of necessary data and discussion. Therefore, my decision of this paper is "major revision". My questions and other sug-

gestions are as follows.

Response: Thank you so much for your patient review and valuable comments. We have tried to improve the manuscript following your comments. Thus, please reconsider the present revision of our manuscript to be published.

1. I suggest add important result of marine MT data acquisition, such as Coherence between horizontal magnetic components and the electric field component, and the Apparent resistivity and impedance phase.

Response: Thanks for your suggestion. The manuscript concentrates on the calibration of the OBEM, its tested procedures, and methodology at the first stage. Therefore, we deployed the first marine OBEM offshore NE Taiwan for testing the hardware, in which the condition of the field was probably not suitable for the marine MT response calculation caused by strong Kuroshio Current passed through the region. Therefore, we would present the marine MT responses after solving all the hardware issues and selecting a suitable region for future deployment at the secondary stage.

2. I suggest add compare with the OBE and OBEM from (kasay, 2009), Qmax3 from Quasar. Add a table present the key specification such as Nosie level, dynamic, power consumption, time drift error size, and cost.

Response: The comparison has added as table 1 and described in lines 60 - 62.

3. The details of electrode, fluxgate sensor and amplifier of your OBEM are not described.

Response: The details of the electrode, fluxgate sensor, and amplifier have described in lines 84 - 90.

4. Figure and tables should be clear and simply. Table1-3 can be replaced with figure. the tables are too large to reduce paper space; figure 1and figure 2 should be redraw to clear; figure4-6, the font size too small to read. Background in Figure 11 is too vague to read.

Response: We have removed all the tables and replaced with the new figures 4 - 6 (pages 35-30)following all your suggestion. The figures 1, 2, and 11 (pages 22, 23 and 33)have also been redrawn in the revised manuscript.

5. The ADS1278 is suitable for audio frequency signal measurement, but the samples is 10Hz, the ADS 1278 is not the best choice. Why choose ADS1278?

Response: We used 8 ch of ADS1278 for issues of multiple input channels, saving circuit volume and the cost of the data logger, whereas the noise level is high through the evaluation. It will be replaced to ADS1281 or others for solving the noise issue in the future. We also mention this as lines 371-373, page 13.

6. The abstract should enhance the innovation design of the OBEM compare the existing OBEM.

Response: We have added the innovative design in the abstract as lines 4-5, page 2. Thanks for your comment.

7. Line 37-line 40 and L45-L49 introduce CSEM method. But your OBEM only used to MT data acquisition. I suggest to delete this sentences.

Response: We have removed all the sentences in the revised manuscript as your suggestion. Thank you.

8. The OBEM integrated the beacon (RF-700A and ST400A) which are expensive. Why not developed a beacon module just like MicrOBS from Sercel?

Response: The reason is that we have used a large amount of these beacons (RF-700A and ST400A) in the broadband OBS system. Thus, we selected the same beacons for the OBEM based on the compatibility and maintenance issues. We are integrating and developing the flash beacon and GPS unit installing into the glass sphere. It would greatly cost down the OBEM system in the future. Thanks for your suggestion.

9. The fluxgate sensor is installed in glass sphere, the distance between data logger and sensor may be very short, how to reduce the disturbance (EM noise) from data logger while writing data to SD?

Response: The distance of the data logger and the magnetic sensor is 45 cm away. The data logger and preamplifiers have been shielded by aluminum foils to GND reducing the EM noise form the data logger. We also separate the battery supplies for the data logger and magnetic sensor, respectively. It could reduce the EM noise from the writing SD card.

10. In 4.1, the description of noise and linearity is too simple to give details. And the test of release is too detailed because the release is from EdgeTech not developed by author.

Response: We have added the noise level of the data logger and its dynamic range in lines 246-247, page 9. The released procedure is very important that would greatly affect the recovery rate of the OBEM. Thus, we would strongly recommend making a detailed of the standard testing procedure although the release hardware made from the EdgeTech instead of the authors.

11. Reference 2 is the same as reference 3?

Response: The reference 2 and reference 3 is different articles. Do you indicate that the references of Chiang et al., 2011 and Chiang et al., 2010 are the same? The Chiang et al., 2011 was a corrigendum of Chiang et al., 2010. If right, we have modified the typing error in the lines 391-393, page 14. Thanks for your comment.

Please also note the supplement to this comment:
https://www.geosci-instrum-method-data-syst-discuss.net/gi-2019-13/gi-2019-13-AC1-supplement.pdf

**Supplement:**

[revised manuscript text omitted]
 which the evaluated results show that the data logger, flush and radio beacons, EMI filter, and an integrated junction board must be improved relating noise levels, cost, and convenient maintenance issues in the future.

**Acknowledgments**

We greatly appreciate the crews of R/V OR2 for the field experiments. The authors acknowledge the financial support from the Ministry of Science and Technology of Taiwan under grant numbers of 105-2116-M-019-001, 106-2116-M-001-008, 106-2116-M-019-003, 107-2116-M-019-006, 108-2116-M-001-012, and 108-2116-M-019-006. We also thank four years of the Taiwan-German cooperative projects on gas hydrate of NEPII for supporting the funds of the instrument deployment of the OBEMs. We would like to thank the TEC Data Center for proving graphical services.

**References**

[revised manuscript text omitted]

**TABLES AND FIGURES**

Table 1

| | Taiwan (OBEM) | Japan (OBEM) | Japan (OBE) |
|---|---|---|---|
| Sampling rate (Hz) | 10 | 8 | 1 |
| AD converter (bits) | 24 | 16 | 24 |
| Resolution (µV/LSB) | 1.5245 | 0.305176 | 0.0019 |
| Resolution of magnetic field (nT/LSB) | 0.010671 | 0.01 | none |
| Max. battery lifetime | About 180 days | About 40 days | About 30 days |
| Power supply | Lithium battery | Lithium battery | Li-ion rechargeable battery |
| Max. memory/Media | 64 GB/ SD card | 2GB/ CF card | 1GB/ CF card |
| Communication port | USB 2.0 | USB1.1/RS-232C | RS-232C |
| Clock drift | < 0.95 ppm | < 2 ppm | < 2 ppm |

Table 2

| Logger S/N | Turn-on Mode (mA) | | | Recording Mode (mA) | | |
|---|---|---|---|---|---|---|
| | 7.2V for Data logger | 7.2V for Sensors | Power consumption | 7.2V for Data logger | 7.2V for Sensors | Power consumption |
| OBEM01 | 32 | 104 | 0.98 | 31 | 105 | 0.98 |
| OBEM02 | 30 | 94 | 0.89 | 29 | 97 | 0.91 |
| OBEM03 | 29 | 103 | 0.95 | 29 | 104 | 0.96 |

Table 3

| | Electrical potential | Impedance | Input DC5V, induce voltage |
|---|---|---|---|
| OBEM01(EX) | 0.56 mV | 245 Ω | 164 mV |
| OBEM01(EY) | 0.26 mV | 272 Ω | 167 mV |
| OBEM02(EX) | 3.63 mV | 243 Ω | 81 mV |
| OBEM02(EY) | 1.93 mV | 370 Ω | 95 mV |
| OBEM03(EX) | 2.38 mV | 267 Ω | 83 mV |
| OBEM03(EY) | 2.1 mV | 331 Ω | 83 mV |

Table 4

| Transducer S/N | Enable Beep (Times) | Disable Beep (Times) | 1st Ranging Distance show on 8011M (m) | 2nd Ranging Distance show on 8011M (m) | 3rd Ranging Distance show on 8011M (m) | 4th Ranging Distance show on 8011M (m) | 5th Ranging Distance show on 8011M (m) | Judgment |
|---|---|---|---|---|---|---|---|---|
| 35427 | 15 | 15 | 629 | 628 | 630 | 627 | 628 | Good |
| 35428 | 15 | 15 | 629 | 627 | 629 | 630 | 629 | Good |
| 35429 | 15 | 15 | 630 | 630 | 630 | 629 | 629 | Good |

**Table 5**

| S/N | Enable Beep (Times) | 1st Ranging Distance show on 8011M (m) | 2nd Ranging Distance show on 8011M (m) | 3rd Ranging Distance show on 8011M (m) | 4th Ranging Distance show on 8011M (m) | 5th Ranging Distance show on 8011M (m) | RELEASE1 Beep Times/Volt | OPTION1 Beep (Times) | RELEASE2 Beep Times/Volt | OPTION1 Beep (Times) | DISABLE Beep (Times) |
|---|---|---|---|---|---|---|---|---|---|---|---|
| 50854 | 15 | 628 | 629 | 630 | 630 | 630 | 15/ 12.77V | 15 | 15/ 12.77V | 15 | 15 |
| 50784 | 7 | 629 | 630 | 630 | 630 | 630 | 7/ 12.77V | 7 | 7/ 12.77V | 7 | 7 |
| 50783 | 15 | 628 | 628 | 628 | 629 | 631 | 15/ 12.77V | 15 | 15/ 12.77V | 15 | 15 |

[Figure]

Figure 1

[Figure]

Figure 2

[Figure]

Figure 3

[Figure]

[Figure]

[Figure]

Figure 4

[Figure]

[Figure]

Figure 5

[Figure]

[Figure]

Figure 6

[Figure]

Figure 7

[Figure]

Figure 8

[Figure]

Figure 9

[Figure]

Figure 10

[Figure]

Figure 11

[Figure]

Figure 12

[Figure]

Figure 13

[Figure]

Figure 14

---

## Referee Comment (RC2) · Denghai Bai (Referee) · 7 Aug 2019

For marine natural resource explorations, the controlled source EM (MCSEM) methods are usually deployed. The natural source EM is used mostly for detecting deep structure of the crust /lithosphere. This manuscript describes a story of evaluations of a natural source marine EM system. I've never seen a paper that gives us so detailed information on how to evaluate an EM instrument. But I'm not sure the work would be published at the moment unless some major revisions be made. Here I'd like to make some suggestions: 1. As I know, some OBEM systems have been developed and deployed in the world (such as SIO-MKIII, LT-OBEM, etc.), I'd like to know what

are the new innovations of the OBEM in this MS? 2. Although the OBEM is a complex integrated system with variety of advanced S&T knowledges, the key units are the data logger, magnetic sensor (fluxgate), electric sensor (electrodes). To evaluate the performance of these delicate units, sophisticated and professional calibrations are needed. The evaluation tools and methods employed in this MS seem like some operational manual step by step which are not enough and unnecessarily described so detailed. For example, the FLUKE726 is an multi-function process calibrator for industrial use, It is not enough for the precise calibration of Geo-EM equipment. 3. For OBEM system the most important feature to be calibrated is the frequency responses in the effective frequency band. I do not find any words about this in the MS. 4. Table 3 to table 7 are really not necessary. Some curved pictures would be better. 5. Calibration of fluxgate is one of the most important work for OBEM. What is the result for this evaluation? 6. Temperature feature is one of key specifications for fluxgate. Calibration of temperature feature should be done before submitting the MS. 7. What does it mean by BBYB and SH1 respectively? 8. It is really strange for the result of the offshore experiment. Why does only the HY component be affected by the earthquakes? 9. Figure 10: Names and notes should be clear and mark the English by the Chinese words. 10. Figure 11: a small inset figure showing the location of the field work should be added. The main body of the figure can use a detailed seafloor topography as the base map.

---

## Author Comment (AC2) · 15 Aug 2019

For marine natural resource explorations, the controlled source EM (MCSEM) methods are usually deployed. The natural source EM is used mostly for detecting deep structure of the crust /lithosphere. This manuscript describes a story of evaluations of a natural source marine EM system. I've never seen a paper that gives us so detailed information on how to evaluate an EM instrument. But I'm not sure the work would be published at the moment unless some major revisions be made. Here I'd like to make some suggestions:

Response: Thank you so much for your patient review and valuable comments. We

have tried to improve the manuscript following your comments. Thus, please re-consider the present revision of our manuscript to be published.

1. As I know, some OBEM systems have been developed and deployed in the world (such as SIO-MKIII, LT-OBEM, etc.), I'd like to know what are the new innovations of the OBEM in this MS?

Response: The innovations of the OBEM have been described in the abstract as lines 2-5 (p. 2). The OBEM can be deployed more than 180 days on the seafloor with time drift less than 0.95 ppm. We also compared with Japanese OBEM in table 1 (p. 19).

2. Although the OBEM is a complex integrated system with variety of advanced S&T knowledges, the key units are the data logger, magnetic sensor (fluxgate), electric sensor (electrodes). To evaluate the performance of these delicate units, sophisticated and professional calibrations are needed. The evaluation tools and methods employed in this MS seem like some operational manual step by step which are not enough and unnecessarily described so detailed. For example, the FLUKE726 is an multi-function process calibrator for industrial use, It is not enough for the precise calibration of Geo-EM equipment.

Response: The MS focuses on the calibration of the OBEM, its tested procedures, and methodology. Therefore, we described the detailed procedures for the OBEMs. All the procedures are very important that would greatly affect the recovery rate of the OBEM. It is impossible to develop the high stability of OBEM without detailed procedures. Thus, we would strongly recommend making detailed testing procedures and manufacturers of equipment. The detailed procedures could be referenced as an operation manual developing OBEM. It is never been described in any articles. Thanks for your comment.

3. For OBEM system the most important feature to be calibrated is the frequency responses in the effective frequency band. I do not find any words about this in the MS.

Response: For the fluxgate, the flat responses show below 10 Hz; the datalogger of the sampling frequency is 144 kHz with linear phase digital filter. Therefore, there is no need to calibrate because of the sampling rate of the datalogger is 10 sps. Thanks for your comment.

4. Table 3 to table 7 are really not necessary. Some curved pictures would be better.

Response: We have replaced the tables 1-3 to be figures 4 - 6 (p. 25-30), whereas the tables 4-7 (tables 2- 5 show in recent MS) probably not suitable making curve pictures because of that show different information of the OBEM. Table 2 shows the very low power consumption of all the OBEMs; table 3 shows the evaluating results of the electrodes; tables 4 and 5 show the evaluation of acoustic transceiver and its transducer in the fieldwork, respectively. All the results demonstrate that the integrated system has properly worked.

5. Calibration of fluxgate is one of the most important work for OBEM. What is the result for this evaluation?

Response: The fluxgate response of amplitude and phase shows in figure 1. Frequency of 0 to 1kHz maximally flat, $\pm5\%$ maximum at 1kHz. The details of the electrode, fluxgate, and amplifier have described in lines 85 - 92.

6. Temperature feature is one of key specifications for fluxgate. Calibration of temperature feature should be done before submitting the MS.

Response: The scaling temperature coefficient is $\pm15$ ppm/$^\circ$C, whereas the offset temperature coefficient is $\pm0.1$ nT/$^\circ$C. We have added the specifications in lines 87-88 (p.4). The temperature for the deep marine environment should change very small compared with the ground case. Therefore, the temperature coefficients of the fluxgate are suitable for the OBEMs.

7. What does it mean by BBYB and SH1 respectively?

Response: BBYB is an abbreviation of "BroadBand YardBird" called for the Taiwanese

OBS as line 46 (p. 3). The SH1 and SH2 are two horizontal components of the seismic signal. We have described it in lines 501 – 503 and lines 507 – 509 (p. 18). Thanks for your valuable comments.

8. It is really strange for the result of the offshore experiment. Why does only the HY component be affected by the earthquakes?

Response: The Hx, Hy and Hz components are simultaneously affected by earthquakes, but the variation of amplitude appears significantly in the Hy component. It could be related to the orientations of the magnetic sensor and the earthquake. We have described the sentences in lines 351 - 353 (p.12). Thanks for your valuable comments.

9. Figure 10: Names and notes should be clear and mark the English by the Chinese words.

Response: We have modified all the Chinese words to English in figure 10 (p. 32). Thanks for your comment.

10. Figure 11: a small inset figure showing the location of the field work should be added. The main body of the figure can use a detailed seafloor topography as the base map.

Response: The figures 11 (p. 33) have been redrawn in the revised MS. Thanks for your comment.

Please also note the supplement to this comment:
https://www.geosci-instrum-method-data-syst-discuss.net/gi-2019-13/gi-2019-13-AC2-supplement.pdf

———————————————

[Figure]

**Fig. 1.** Response of fluxgate

**Supplement:**

[revised manuscript text omitted]

and SH2: two horizontal components of the seismic signal; SZ: vertical component of the seismic signal.

Figure 14. The variations in PGV, TMF, HY, TX, and TY during the first earthquake.

The PGV of 2.63 cm/s affected the inclinations by 0.601° and 0.404° for TX and TY, respectively, and the HY magnetic field had a peak of 100 nT. SH1 and SH2: two horizontal components of the seismic signal; SZ: vertical component of the seismic signal.

**TABLES AND FIGURES**

Table 1

| | Taiwan (OBEM) | Japan (OBEM) | Japan (OBE) |
|---|---|---|---|
| Sampling rate (Hz) | 10 | 8 | 1 |
| AD converter (bits) | 24 | 16 | 24 |
| Resolution (μV/LSB) | 1.5245 | 0.305176 | 0.0019 |
| Resolution of magnetic field (nT/LSB) | 0.010671 | 0.01 | none |
| Max. battery lifetime | About 180 days | About 40 days | About 30 days |
| Power supply | Lithium battery | Lithium battery | Li-ion rechargeable battery |
| Max. memory/Media | 64 GB/ SD card | 2GB/ CF card | 1GB/ CF card |
| Communication port | USB 2.0 | USB1.1/RS-232C | RS-232C |
| Clock drift | < 0.95 ppm | < 2 ppm | < 2 ppm |

Table 2

| Logger S/N | Turn-on Mode (mA) | | | Recording Mode (mA) | | |
|---|---|---|---|---|---|---|
| | 7.2V for Data logger | 7.2V for Sensors | Power consumption | 7.2V for Data logger | 7.2V for Sensors | Power consumption |
| OBEM01 | 32 | 104 | 0.98 | 31 | 105 | 0.98 |
| OBEM02 | 30 | 94 | 0.89 | 29 | 97 | 0.91 |
| OBEM03 | 29 | 103 | 0.95 | 29 | 104 | 0.96 |

Table 3

| | Electrical potential | Impedance | Input DC5V, induce voltage |
|---|---|---|---|
| OBEM01(EX) | 0.56 mV | 245 Ω | 164 mV |
| OBEM01(EY) | 0.26 mV | 272 Ω | 167 mV |
| OBEM02(EX) | 3.63 mV | 243 Ω | 81 mV |
| OBEM02(EY) | 1.93 mV | 370 Ω | 95 mV |
| OBEM03(EX) | 2.38 mV | 267 Ω | 83 mV |
| OBEM03(EY) | 2.1 mV | 331 Ω | 83 mV |

Table 4

| Transducer S/N | Enable Beep (Times) | Disable Beep (Times) | 1st Ranging Distance show on 8011M (m) | 2nd Ranging Distance show on 8011M (m) | 3rd Ranging Distance show on 8011M (m) | 4th Ranging Distance show on 8011M (m) | 5th Ranging Distance show on 8011M (m) | Judgment |
|---|---|---|---|---|---|---|---|---|
| 35427 | 15 | 15 | 629 | 628 | 630 | 627 | 628 | Good |
| 35428 | 15 | 15 | 629 | 627 | 629 | 630 | 629 | Good |
| 35429 | 15 | 15 | 630 | 630 | 630 | 629 | 629 | Good |

**Table 5**

| S/N | Enable Beep (Times) | 1st Ranging Distance show on 8011M (m) | 2nd Ranging Distance show on 8011M (m) | 3rd Ranging Distance show on 8011M (m) | 4th Ranging Distance show on 8011M (m) | 5th Ranging Distance show on 8011M (m) | RELEASE1 Beep Times/Volt | OPTION1 Beep (Times) | RELEASE2 Beep Times/Volt | OPTION1 Beep (Times) | DISABLE Beep (Times) |
|---|---|---|---|---|---|---|---|---|---|---|---|
| 50854 | 15 | 628 | 629 | 630 | 630 | 630 | 15/ 12.77V | 15 | 15/ 12.77V | 15 | 15 |
| 50784 | 7 | 629 | 630 | 630 | 630 | 630 | 7/ 12.77V | 7 | 7/ 12.77V | 7 | 7 |
| 50783 | 15 | 628 | 628 | 628 | 629 | 631 | 15/ 12.77V | 15 | 15/ 12.77V | 15 | 15 |

[Figure]

Figure 1

[Figure]

Figure 2

[Figure]

Figure 3

[Figure]

[Figure]

Figure 4

[Figure]

[Figure]

Figure 5

[Figure]

[Figure]

Figure 6

[Figure]

Figure 7

[Figure]

Figure 8

[Figure]

Figure 9

[Figure]

Figure 10

[Figure]

Figure 11

[Figure]

Figure 12

[Figure]

Figure 13

[Figure]

Figure 14